# Fast Second-Order Stochastic Backpropagation for Variational Inference

**Kai Fan**
Duke University
kai.fan@stat.duke.edu

**Ziteng Wang**[*]
HKUST[†]
wangzt2012@gmail.com

**Jeffrey Beck**
Duke University
jeff.beck@duke.edu

**James T. Kwok**
HKUST
jamesk@cse.ust.hk

**Katherine Heller**
Duke University
kheller@gmail.com

## Abstract

We propose a second-order (Hessian or Hessian-free) based optimization method for variational inference inspired by Gaussian backpropagation, and argue that quasi-Newton optimization can be developed as well. This is accomplished by generalizing the gradient computation in stochastic backpropagation via a reparametrization trick with lower complexity. As an illustrative example, we apply this approach to the problems of Bayesian logistic regression and variational auto-encoder (VAE). Additionally, we compute bounds on the estimator variance of intractable expectations for the family of Lipschitz continuous function. Our method is practical, scalable and model free. We demonstrate our method on several real-world datasets and provide comparisons with other stochastic gradient methods to show substantial enhancement in convergence rates.

## 1 Introduction

Generative models have become ubiquitous in machine learning and statistics and are now widely used in fields such as bioinformatics, computer vision, or natural language processing. These models benefit from being highly interpretable and easily extended. Unfortunately, inference and learning with generative models is often intractable, especially for models that employ continuous latent variables, and so fast approximate methods are needed. Variational Bayesian (VB) methods [1] deal with this problem by approximating the true posterior that has a tractable parametric form and then identifying the set of parameters that maximize a variational lower bound on the marginal likelihood. That is, VB methods turn an inference problem into an optimization problem that can be solved, for example, by gradient ascent.

Indeed, efficient stochastic gradient variational Bayesian (SGVB) estimators have been developed for auto-encoder models [17] and a number of papers have followed up on this approach [28, 25, 19, 16, 15, 26, 10]. Recently, [25] provided a complementary perspective by using stochastic backpropagation that is equivalent to SGVB and applied it to deep latent gaussian models. Stochastic backpropagation overcomes many limitations of traditional inference methods such as the mean-field or wake-sleep algorithms [12] due to the existence of efficient computations of an unbiased estimate of the gradient of the variational lower bound. The resulting gradients can be used for parameter estimation via stochastic optimization methods such as stochastic gradient decent(SGD) or adaptive version (Adagrad) [6].

---

[*]Equal Contribution to this work
[†]Refer to Hong Kong University of Science and Technology

Unfortunately, methods such as SGD or Adagrad converge slowly for some difficult-to-train models, such as untied-weights auto-encoders or recurrent neural networks. The common experience is that gradient decent always gets stuck near saddle points or local extrema. Meanwhile the learning rate is difficult to tune. [18] gave a clear explanation on why Newton's method is preferred over gradient decent, which often encounters under-fitting problem if the optimizing function manifests pathological curvature. Newton's method is invariant to affine transformations so it can take advantage of curvature information, but has higher computational cost due to its reliance on the inverse of the Hessian matrix. This issue was partially addressed in [18] where the authors introduced Hessian free (HF) optimization and demonstrated its suitability for problems in machine learning.

In this paper, we continue this line of research into 2[nd] order variational inference algorithms. Inspired by the property of location scale families [8], we show how to reduce the computational cost of the Hessian or Hessian-vector product, thus allowing for a 2[nd] order stochastic optimization scheme for variational inference under Gaussian approximation. In conjunction with the HF optimization, we propose an efficient and scalable 2[nd] order stochastic Gaussian backpropagation for variational inference called HFSGVI. Alternately, L-BFGS [3] version, a quasi-Newton method merely using the gradient information, is a natural generalization of 1[st] order variational inference.

The most immediate application would be to look at obtaining better optimization algorithms for variational inference. As to our knowledge, the model currently applying 2[nd] order information is LDA [2, 14], where the Hessian is easy to compute [11]. In general, for non-linear factor models like non-linear factor analysis or the deep latent Gaussian models this is not the case. Indeed, to our knowledge, there has not been any systematic investigation into the properties of various optimization algorithms and how they might impact the solutions to optimization problem arising from variational approximations.

The main contributions of this paper are to fill such gap for variational inference by introducing a novel 2[nd] order optimization scheme. First, we describe a clever approach to obtain curvature information with low computational cost, thus making the Newton's method both scalable and efficient. Second, we show that the variance of the lower bound estimator can be bounded by a dimension-free constant, extending the work of [25] that discussed a specific bound for univariate function. Third, we demonstrate the performance of our method for Bayesian logistic regression and the VAE model in comparison to commonly used algorithms. Convergence rate is shown to be competitive or faster.

## 2 Stochastic Backpropagation

In this section, we extend the Bonnet and Price theorem [4, 24] to develop 2[nd] order Gaussian backpropagation. Specifically, we consider how to optimize an expectation of the form $\mathbb{E}_{q_\theta}[f(\mathbf{z}|\mathbf{x})]$, where $\mathbf{z}$ and $\mathbf{x}$ refer to latent variables and observed variables respectively, and expectation is taken w.r.t distribution $q_\theta$ and $f$ is some smooth loss function (e.g. it can be derived from a standard variational lower bound [1]). Sometimes we abuse notation and refer to $f(\mathbf{z})$ by omitting $\mathbf{x}$ when no ambiguity exists. To optimize such expectation, gradient decent methods require the 1[st] derivatives, while Newton's methods require both the gradients and Hessian involving 2[nd] order derivatives.

### 2.1 Second Order Gaussian Backpropagation

If the distribution $q$ is a $d_z$-dimensional Gaussian $\mathcal{N}(\mathbf{z}|\boldsymbol{\mu}, \mathbf{C})$, the required partial derivative is easily computed with a lower algorithmic cost $\mathcal{O}(d_z^2)$ [25]. By using the property of Gaussian distribution, we can compute the 2[nd] order partial derivative of $\mathbb{E}_{\mathcal{N}(\mathbf{z}|\boldsymbol{\mu}, \mathbf{C})}[f(\mathbf{z})]$ as follows:

$$\nabla^2_{\mu_i, \mu_j} \mathbb{E}_{\mathcal{N}(\mathbf{z}|\boldsymbol{\mu}, \mathbf{C})}[f(\mathbf{z})] \quad = \quad \mathbb{E}_{\mathcal{N}(\mathbf{z}|\boldsymbol{\mu}, \mathbf{C})}[\nabla^2_{z_i, z_j} f(\mathbf{z})] = 2\nabla_{C_{ij}} \mathbb{E}_{\mathcal{N}(\mathbf{z}|\boldsymbol{\mu}, \mathbf{C})}[f(\mathbf{z})], \quad (1)$$

$$\nabla^2_{C_{i,j}, C_{k,l}} \mathbb{E}_{\mathcal{N}(\mathbf{z}|\boldsymbol{\mu}, \mathbf{C})}[f(\mathbf{z})] \quad = \quad \frac{1}{4}\mathbb{E}_{\mathcal{N}(\mathbf{z}|\boldsymbol{\mu}, \mathbf{C})}[\nabla^4_{z_i, z_j, z_k, z_l} f(\mathbf{z})], \quad (2)$$

$$\nabla^2_{\mu_i, C_{k,l}} \mathbb{E}_{\mathcal{N}(\mathbf{z}|\boldsymbol{\mu}, \mathbf{C})}[f(\mathbf{z})] \quad = \quad \frac{1}{2}\mathbb{E}_{\mathcal{N}(\mathbf{z}|\boldsymbol{\mu}, \mathbf{C})}\left[\nabla^3_{z_i, z_k, z_l} f(\mathbf{z})\right]. \quad (3)$$

Eq. (1), (2), (3) (proof in supplementary) have the nice property that a limited number of samples from $q$ are sufficient to obtain unbiased gradient estimates. However, note that Eq. (2), (3) needs to calculate the third and fourth derivatives of $f(\mathbf{z})$, which is highly computationally inefficient. To avoid the calculation of high order derivatives, we use a co-ordinate transformation.

## 2.2 Covariance Parameterization for Optimization

By constructing the linear transformation (a.k.a. reparameterization) $\mathbf{z} = \boldsymbol{\mu} + \mathbf{R}\boldsymbol{\epsilon}$, where $\boldsymbol{\epsilon} \sim \mathcal{N}(0, \mathbf{I}_{d_z})$, we can generate samples from any Gaussian distribution $\mathcal{N}(\boldsymbol{\mu}, \mathbf{C})$ by simulating data from a standard normal distribution, provided the decomposition $\mathbf{C} = \mathbf{R}\mathbf{R}^\top$ holds. This fact allows us to derive the following theorem indicating that the computation of $2^{\text{nd}}$ order derivatives can be scalable and programmed to run in parallel.

**Theorem 1** (**Fast Derivative**). *If $f$ is a twice differentiable function and $\mathbf{z}$ follows Gaussian distribution $\mathcal{N}(\boldsymbol{\mu}, \mathbf{C})$, $\mathbf{C} = \mathbf{R}\mathbf{R}^\top$, where both the mean $\boldsymbol{\mu}$ and $\mathbf{R}$ depend on a $d$-dimensional parameter $\boldsymbol{\theta} = (\theta_l)_{l=1}^d$, i.e. $\boldsymbol{\mu}(\boldsymbol{\theta}), \mathbf{R}(\boldsymbol{\theta})$, we have $\nabla_{\boldsymbol{\mu}, \mathbf{R}}^2 \mathbb{E}_{\mathcal{N}(\boldsymbol{\mu}, \mathbf{C})}[f(\mathbf{z})] = \mathbb{E}_{\boldsymbol{\epsilon} \sim \mathcal{N}(0, \mathbf{I}_{d_z})}[\boldsymbol{\epsilon}^\top \otimes \mathbf{H}]$, and $\nabla_{\mathbf{R}}^2 \mathbb{E}_{\mathcal{N}(\boldsymbol{\mu}, \mathbf{C})}[f(\mathbf{z})] = \mathbb{E}_{\boldsymbol{\epsilon} \sim \mathcal{N}(0, \mathbf{I}_{d_z})}[(\boldsymbol{\epsilon}\boldsymbol{\epsilon}^T) \otimes \mathbf{H}]$. This then implies*

$$\nabla_{\theta_l} \mathbb{E}_{\mathcal{N}(\boldsymbol{\mu}, \mathbf{C})}[f(\mathbf{z})] = \mathbb{E}_{\boldsymbol{\epsilon} \sim \mathcal{N}(0, \mathbf{I})} \left[ \mathbf{g}^\top \frac{\partial(\boldsymbol{\mu} + \mathbf{R}\boldsymbol{\epsilon})}{\partial \theta_l} \right], \tag{4}$$

$$\nabla_{\theta_{l_1} \theta_{l_2}}^2 \mathbb{E}_{\mathcal{N}(\boldsymbol{\mu}, \mathbf{C})}[f(\mathbf{z})] = \mathbb{E}_{\boldsymbol{\epsilon} \sim \mathcal{N}(0, \mathbf{I})} \left[ \frac{\partial(\boldsymbol{\mu} + \mathbf{R}\boldsymbol{\epsilon})}{\partial \theta_{l_1}}^\top \mathbf{H} \frac{\partial(\boldsymbol{\mu} + \mathbf{R}\boldsymbol{\epsilon})}{\partial \theta_{l_2}} + \mathbf{g}^\top \frac{\partial^2(\boldsymbol{\mu} + \mathbf{R}\boldsymbol{\epsilon})}{\partial \theta_{l_1} \partial \theta_{l_2}} \right], \tag{5}$$

*where $\otimes$ is Kronecker product, and gradient $\mathbf{g}$, Hessian $\mathbf{H}$ are evaluated at $\boldsymbol{\mu} + \mathbf{R}\boldsymbol{\epsilon}$ in terms of $f(\mathbf{z})$.*

If we consider the mean and covariance matrix as the variational parameters in variational inference, the first two results w.r.t $\boldsymbol{\mu}, \mathbf{R}$ make parallelization possible, and reduce computational cost of the Hessian-vector multiplication due to the fact that $(A^\top \otimes B)vec(V) = vec(AVB)$. If the model has few parameters or a large resource budget (e.g. GPU) is allowed, Theorem 1 launches the foundation for exact $2^{\text{nd}}$ order derivative computation in parallel. In addition, note that the $2^{\text{nd}}$ order gradient computation on model parameter $\boldsymbol{\theta}$ only involves matrix-vector or vector-vector multiplication, thus leading to an algorithmic complexity that is $\mathcal{O}(d_z^2)$ for $2^{\text{nd}}$ order derivative of $\boldsymbol{\theta}$, which is the same as $1^{\text{st}}$ order gradient [25]. The derivative computation at function $f$ is up to $2^{\text{nd}}$ order, avoiding to calculate $3^{\text{rd}}$ or $4^{\text{th}}$ order derivatives. One practical parametrization assumes a diagonal covariance matrix $\mathbf{C} = \text{diag}\{\sigma_1^2, ..., \sigma_{d_z}^2\}$. This reduces the actual computational cost compared with Theorem 1, albeit the same order of the complexity ($\mathcal{O}(d_z^2)$) (see supplementary material). Theorem 1 holds for a large class of distributions in addition to Gaussian distributions, such as student $t$-distribution. If the dimensionality $d$ of embedded parameter $\boldsymbol{\theta}$ is large, computation of the gradient $\mathbf{G}_{\boldsymbol{\theta}}$ and Hessian $\mathbf{H}_{\boldsymbol{\theta}}$ (differ from $\mathbf{g}, \mathbf{H}$ above) will be linear and quadratic w.r.t $d$, which may be unacceptable. Therefore, in the next section we attempt to reduce the computational complexity w.r.t $d$.

## 2.3 Apply Reparameterization on Second Order Algorithm

In standard Newton's method, we need to compute the Hessian matrix and its inverse, which is intractable for limited computing resources. [18] applied Hessian-free (HF) optimization method in deep learning effectively and efficiently. This work largely relied on the technique of fast Hessian matrix-vector multiplication [23]. We combine reparameterization trick with Hessian-free or quasi-Newton to circumvent matrix inverse problem.

**Hessian-free** Unlike quasi-Newton methods HF doesn't make any approximation on the Hessian. HF needs to compute $\mathbf{H}_{\boldsymbol{\theta}}\mathbf{v}$, where $\mathbf{v}$ is any vector that has the matched dimension to $\mathbf{H}_{\boldsymbol{\theta}}$, and then uses conjugate gradient algorithm to solve the linear system $\mathbf{H}_{\boldsymbol{\theta}}\mathbf{v} = -\nabla F(\boldsymbol{\theta})^\top \mathbf{v}$, for any objective function $F$. [18] gives a reasonable explanation for Hessian free optimization. In short, unlike a pre-training method that places the parameters in a search region to regularize[7], HF solves issues of pathological curvature in the objective by taking the advantage of rescaling property of Newton's method. By definition $\mathbf{H}_{\boldsymbol{\theta}}\mathbf{v} = \lim_{\gamma \to 0} \frac{\nabla F(\boldsymbol{\theta} + \gamma \mathbf{v}) - \nabla F(\boldsymbol{\theta})}{\gamma}$ indicating that $\mathbf{H}_{\boldsymbol{\theta}}\mathbf{v}$ can be numerically computed by using finite differences at $\gamma$. However, this numerical method is unstable for small $\gamma$.

In this section, we focus on the calculation of $\mathbf{H}_{\boldsymbol{\theta}}\mathbf{v}$ by leveraging a reparameterization trick. Specifically, we apply an $\mathcal{R}$-operator technique [23] for computing the product $\mathbf{H}_{\boldsymbol{\theta}}\mathbf{v}$ exactly. Let $F = \mathbb{E}_{\mathcal{N}(\boldsymbol{\mu}, \mathbf{C})}[f(\mathbf{z})]$ and reparametrize $\mathbf{z}$ again as Sec. 2.2, we do variable substitution $\boldsymbol{\theta} \leftarrow \boldsymbol{\theta} + \gamma \mathbf{v}$ after gradient Eq. (4) is obtained, and then take derivative on $\gamma$. Thus we have the following analyt-

---

**Algorithm 1** Hessian-free Algorithm on Stochastic Gaussian Variational Inference (HFSGVI)

---

**Parameters:** Minibatch Size $B$, Number of samples to estimate the expectation $M$ ($= 1$ as default),
**Input:** Observation $\mathbf{X}$ (and $\mathbf{Y}$ if required), Lower bound function $\mathcal{L} = \mathbb{E}_{\mathcal{N}(\boldsymbol{\mu}, \mathbf{C})}[f_{\mathcal{L}}]$
**Output:** Parameter $\boldsymbol{\theta}$ after having converged.

1: **for** $t = 1, 2, \ldots$ **do**
2:     $\mathbf{x}_{b=1}^{B} \leftarrow$ Randomly draw $B$ datapoints from full data set $\mathbf{X}$;
3:     $\boldsymbol{\epsilon}_{m_b=1}^{M} \leftarrow$ sample $M$ times from $\mathcal{N}(0, \mathbf{I})$ for each $\mathbf{x}_b$;
4:     Define gradient $\mathbf{G}(\boldsymbol{\theta}) = \frac{1}{M} \sum_b \sum_{m_b} \mathbf{g}_{b,m}^{\top} \frac{\partial(\boldsymbol{\mu} + \mathbf{R}\boldsymbol{\epsilon}_{m_b})}{\partial\boldsymbol{\theta}}, \mathbf{g}_{b,m} = \nabla_{\mathbf{z}}(f_{\mathcal{L}}(\mathbf{z}|\mathbf{x}_b))|_{\mathbf{z} = \boldsymbol{\mu} + \mathbf{R}\boldsymbol{\epsilon}_{m_b}}$;
5:     Define function $\mathbf{B}(\boldsymbol{\theta}, \mathbf{v}) = \nabla_{\gamma} \mathbf{G}(\boldsymbol{\theta} + \gamma\mathbf{v})|_{\gamma=0}$, where $\mathbf{v}$ is a $d$-dimensional vector;
6:     Using Conjugate Gradient algorithm to solve linear system: $\mathbf{B}(\boldsymbol{\theta}_t, \mathbf{p}_t) = -\mathbf{G}(\boldsymbol{\theta}_t)$;
7:     $\boldsymbol{\theta}_{t+1} = \boldsymbol{\theta}_t + \mathbf{p}_t$;
8: **end for**

---

ical expression for Hessian-vector multiplication:

$$
\begin{aligned}
\mathbf{H}_{\boldsymbol{\theta}}\mathbf{v} &= \left. \frac{\partial}{\partial\gamma} \nabla F(\boldsymbol{\theta} + \gamma\mathbf{v}) \right|_{\gamma=0} = \frac{\partial}{\partial\gamma} \mathbb{E}_{\mathcal{N}(0,\mathbf{I})} \left[ \mathbf{g}^{\top} \left. \frac{\partial(\boldsymbol{\mu}(\boldsymbol{\theta}) + \mathbf{R}(\boldsymbol{\theta})\boldsymbol{\epsilon})}{\partial\boldsymbol{\theta}} \right|_{\boldsymbol{\theta} \leftarrow \boldsymbol{\theta} + \gamma\mathbf{v}} \right] \Bigg|_{\gamma=0} \\
&= \mathbb{E}_{\mathcal{N}(0,\mathbf{I})} \left[ \frac{\partial}{\partial\gamma} \left( \mathbf{g}^{\top} \left. \frac{\partial(\boldsymbol{\mu}(\boldsymbol{\theta}) + \mathbf{R}(\boldsymbol{\theta})\boldsymbol{\epsilon})}{\partial\boldsymbol{\theta}} \right|_{\boldsymbol{\theta} \leftarrow \boldsymbol{\theta} + \gamma\mathbf{v}} \right) \right] \Bigg|_{\gamma=0}. \quad (6)
\end{aligned}
$$

Eq. (6) is appealing since it does not need to store the dense matrix and provides an unbiased $\mathbf{H}_{\boldsymbol{\theta}}\mathbf{v}$ estimator with a small sample size. In order to conduct the 2$^{\text{nd}}$ order optimization for variational inference, if the computation of the gradient for variational lower bound is completed, we only need to add one extra step for gradient evaluation via Eq. (6) which has the same computational complexity as Eq. (4). This leads to a Hessian-free variational inference method described in Algorithm 1.

For the worst case of HF, the conjugate gradient (CG) algorithm requires at most $d$ iterations to terminate, meaning that it requires $d$ evaluations of $\mathbf{H}_{\boldsymbol{\theta}}\mathbf{v}$ product. However, the good news is that CG leads to good convergence after a reasonable number of iterations. In practice we found that it may not necessary to wait CG to converge. In other words, even if we set the maximum iteration $K$ in CG to a small fixed number (e.g., 10 in our experiments, though with thousands of parameters), the performance does not deteriorate. The early stoping strategy may have the similar effect of Wolfe condition to avoid excessive step size in Newton's method. Therefore we successfully reduce the complexity of each iteration to $\mathcal{O}(Kdd_z^2)$, whereas $\mathcal{O}(dd_z^2)$ is for one SGD iteration.

**L-BFGS** Limited memory BFGS utilizes the information gleaned from the gradient vector to approximate the Hessian matrix without explicit computation, and we can readily utilize it within our framework. The basic idea of BFGS approximates Hessian by an iterative algorithm $\mathbf{B}_{t+1} = \mathbf{B}_t + \Delta\mathbf{G}_t\Delta\mathbf{G}_t^{\top}/\Delta\boldsymbol{\theta}_t\Delta\boldsymbol{\theta}_t^{\top} - \mathbf{B}_t\Delta\boldsymbol{\theta}_t\Delta\boldsymbol{\theta}_t^{\top}\mathbf{B}_t/\Delta\boldsymbol{\theta}_t^{\top}\mathbf{B}_t\Delta\boldsymbol{\theta}_t$, where $\Delta\mathbf{G}_t = \mathbf{G}_t - \mathbf{G}_{t-1}$ and $\Delta\boldsymbol{\theta}_t = \boldsymbol{\theta}_t - \boldsymbol{\theta}_{t-1}$. By Eq. (4), the gradient $\mathbf{G}_t$ at each iteration can be obtained without any difficulty. However, even if this low rank approximation to the Hessian is easy to invert analytically due to the Sherman-Morrison formula, we still need to store the matrix. L-BFGS will further implicitly approximate this dense $\mathbf{B}_t$ or $\mathbf{B}_t^{-1}$ by tracking only a few gradient vectors and a short history of parameters and therefore has a linear memory requirement. In general, L-BFGS can perform a sequence of inner products with the $K$ most recent $\Delta\boldsymbol{\theta}_t$ and $\Delta\mathbf{G}_t$, where $K$ is a predefined constant (10 or 15 in our experiments). Due to the space limitations, we omit the details here but none-the-less will present this algorithm in experiments section.

## 2.4 Estimator Variance

The framework of stochastic backpropagation [16, 17, 19, 25] extensively uses the mean of very few samples (often just one) to approximate the expectation. Similarly we approximate the left side of Eq. (4), (5), (6) by sampling few points from the standard normal distribution. However, the magnitude of the variance of such an estimator is not seriously discussed. [25] simply explored the variance quantitatively for separable functions.[19] merely borrowed the variance reduction technique from reinforcement learning by centering the learning signal in expectation and performing variance normalization. Here, we will generalize the treatment of variance to a broader family, Lipschitz continuous function.

**Theorem 2** (**Variance Bound**). *If $f$ is an $L$-Lipschitz differentiable function and $\epsilon \sim \mathcal{N}(0, \mathbf{I}_{d_z})$, then $\mathbb{E}[(f(\epsilon) - \mathbb{E}[f(\epsilon)])^2] \leq \frac{L^2 \pi^2}{4}$.*

The proof of Theorem 2 (see supplementary) employs the properties of sub-Gaussian distributions and the duplication trick that are commonly used in learning theory. Significantly, the result implies a variance bound independent of the dimensionality of Gaussian variable. Note that from the proof, we can only obtain the $\mathbb{E}\left[e^{\lambda(f(\epsilon) - \mathbb{E}[f(\epsilon)])}\right] \leq e^{L^2 \lambda^2 \pi^2 / 8}$ for $\lambda > 0$. Though this result is enough to illustrate the variance independence of $d_z$, we can in fact tighten it to a sharper upper bound by a constant scalar, i.e. $e^{\lambda^2 L^2 / 2}$, thus leading to the result of Theorem 2 with $\text{Var}(f(\epsilon)) \leq L^2$. If all the results above hold for smooth (twice continuous and differentiable) functions with Lipschitz constant $L$ then it holds for all Lipschitz functions by a standard approximation argument. This means the condition can be relaxed to Lipschitz continuous function.

**Corollary 3** (**Bias Bound**). $\mathbb{P}\left( \left| \frac{1}{M} \sum_{m=1}^{M} f(\epsilon_m) - \mathbb{E}[f(\epsilon)] \right| \geq t \right) \leq 2 e^{-\frac{2Mt^2}{\pi^2 L^2}}$ .

It is also worth mentioning that the significant corollary of Theorem 2 is probabilistic inequality to measure the convergence rate of Monte Carlo approximation in our setting. This tail bound, together with variance bound, provides the theoretical guarantee for stochastic backpropagation on Gaussian variables and provides an explanation for why a unique realization ($M = 1$) is enough in practice. By reparametrization, Eq. (4), (5, (6) can be formulated as the expectation w.r.t the isotropic Gaussian distribution with identity covariance matrix leading to Algorithm 1. Thus we can rein in the number of samples for Monte Carlo integration regardless dimensionality of latent variables $\mathbf{z}$. This seems counter-intuitive. However, we notice that larger $L$ may require more samples, and Lipschitz constants of different models vary greatly.

## 3 Application on Variational Auto-encoder

Note that our method is model free. If the loss function has the form of the expectation of a function w.r.t latent Gaussian variables, we can directly use Algorithm 1. In this section, we put the emphasis on a standard framework VAE model [17] that has been intensively researched; in particular, the function endows the logarithm form, thus bridging the gap between Hessian and fisher information matrix by expectation (see a survey [22] and reference therein).

### 3.1 Model Description

Suppose we have $N$ i.i.d. observations $\mathbf{X} = \{\mathbf{x}^{(i)}\}_{i=1}^{N}$, where $\mathbf{x}^{(i)} \in \mathbb{R}^D$ is a data vector that can take either continuous or discrete values. In contrast to a standard auto-encoder model constructed by a neural network with a bottleneck structure, VAE describes the embedding process from the prospective of a Gaussian latent variable model. Specifically, each data point $\mathbf{x}$ follows a generative model $p_{\psi}(\mathbf{x}|\mathbf{z})$, where this process is actually a decoder that is usually constructed by a non-linear transformation with unknown parameters $\psi$ and a prior distribution $p_{\psi}(\mathbf{z})$. The encoder or recognition model $q_{\phi}(\mathbf{z}|\mathbf{x})$ is used to approximate the true posterior $p_{\psi}(\mathbf{z}|\mathbf{x})$, where $\phi$ is similar to the parameter of variational distribution. As suggested in [16, 17, 25], multi-layered perceptron (MLP) is commonly considered as both the probabilistic encoder and decoder. We will later see that this construction is equivalent to a variant deep neural networks under the constrain of unique realization for $\mathbf{z}$. For this model and each datapoint, the variational lower bound on the marginal likelihood is,

$$\log p_{\psi}(\mathbf{x}^{(i)}) \geq \mathbb{E}_{q_{\phi}(\mathbf{z}|\mathbf{x}^{(i)})}[\log p_{\psi}(\mathbf{x}^{(i)}|\mathbf{z})] - D_{KL}(q_{\phi}(\mathbf{z}|\mathbf{x}^{(i)})\|p_{\psi}(\mathbf{z})) = \mathcal{L}(\mathbf{x}^{(i)}). \quad (7)$$

We can actually write the KL divergence into the expectation term and denote $(\psi, \phi)$ as $\boldsymbol{\theta}$. By the previous discussion, this means that our objective is to solve the optimization problem $\arg\max_{\boldsymbol{\theta}} \sum_i \mathcal{L}(\mathbf{x}^{(i)})$ of full dataset variational lower bound. Thus L-BFGS or HF SGVI algorithm can be implemented straightforwardly to estimate the parameters of both generative and recognition models. Since the first term of reconstruction error appears in Eq. (7) with an expectation form on latent variable, [17, 25] used a small finite number $M$ samples as Monte Carlo integration with reparameterization trick to reduce the variance. This is, in fact, drawing samples from the standard normal distribution. In addition, the second term is the KL divergence between the variational distribution and the prior distribution, which acts as a regularizer.

### 3.2 Deep Neural Networks with Hybrid Hidden Layers

In the experiments, setting $M = 1$ can not only achieve excellent performance but also speed up the program. In this special case, we discuss the relationship between VAE and traditional deep auto-encoder. For binary inputs, denote the output as $\mathbf{y}$, we have $\log p_{\psi}(\mathbf{x}|\mathbf{z}) = \sum_{j=1}^{D} x_j \log y_j + (1 - x_j) \log(1 - y_j)$, which is exactly the negative cross-entropy. It is also apparent that $\log p_{\psi}(\mathbf{x}|\mathbf{z})$ is equivalent to negative squared error loss for continuous data. This means that maximizing the lower bound is roughly equal to minimizing the loss function of a deep neural network (see Figure 1 in supplementary), except for different regularizers. In other words, the prior in VAE only imposes a regularizer in encoder or generative model, while $\mathcal{L}_2$ penalty for all parameters is always considered in deep neural nets. From the perspective of deep neural networks with hybrid hidden nodes, the model consists of two Bernoulli layers and one Gaussian layer. The gradient computation can simply follow a variant of backpropagation layer by layer (derivation given in supplementary). To further see the rationale of setting $M = 1$, we will investigate the upper bound of the Lipschitz constant under various activation functions in the next lemma. As Theorem 2 implies, the variance of approximate expectation by finite samples mainly relies on the Lipschitz constant, rather than dimensionality. According to Lemma 4, imposing a prior or regularization to the parameter can control both the model complexity and function smoothness. Lemma 4 also implies that we can get the upper bound of the Lipschitz constant for the designed estimators in our algorithm.

**Lemma 4.** *For a sigmoid activation function $g$ in deep neural networks with one Gaussian layer $\mathbf{z}$, $\mathbf{z} \sim \mathcal{N}(\boldsymbol{\mu}, \mathbf{C}), \mathbf{C} = \mathbf{R}^{\top}\mathbf{R}$. Let $\mathbf{z} = \boldsymbol{\mu} + \mathbf{R}\boldsymbol{\epsilon}$, then the Lipschitz constant of $g(W_{i,}(\boldsymbol{\mu} + \mathbf{R}\boldsymbol{\epsilon}) + b_i)$ is bounded by $\frac{1}{4}\|W_{i,}\mathbf{R}\|_2$, where $W_{i,}$ is ith row of weight matrix and $b_i$ is the ith element bias. Similarly, for hyperbolic tangent or softplus function, the Lipschitz constant is bounded by $\|W_{i,}\mathbf{R}\|_2$.*

## 4 Experiments

We apply our $2^{\text{nd}}$ order stochastic variational inference to two different non-conjugate models. First, we consider a simple but widely used Bayesian logistic regression model, and compare with the most recent $1^{\text{st}}$ order algorithm, doubly stochastic variational inference (DSVI) [28], designed for sparse variable selection with logistic regression. Then, we compare the performance of VAE model with our algorithms.

### 4.1 Bayesian Logistic Regression

Given a dataset $\{\mathbf{x}_i, y_i\}_{i=1}^{N}$, where each instance $\mathbf{x}_i \in \mathbb{R}^D$ includes the default feature 1 and $y_i \in \{-1, 1\}$ is the binary label, the Bayesian logistic regression models the probability of outputs conditional on features and the coefficients $\boldsymbol{\beta}$ with an imposed prior. The likelihood and the prior can usually take the form as $\prod_{i=1}^{N} g(y_i \mathbf{x}_i^{\top} \boldsymbol{\beta})$ and $\mathcal{N}(0, \boldsymbol{\Lambda})$ respectively, where $g$ is sigmoid function and $\boldsymbol{\Lambda}$ is a diagonal covariance matrix for simplicity. We can propose a variational Gaussian distribution $q(\boldsymbol{\beta}|\boldsymbol{\mu}, \mathbf{C})$ to approximate the posterior of regression parameter. If we further assume a diagonal $\mathbf{C}$, a factorized form $\prod_{j=1}^{D} q(\beta_j|\mu_j, \sigma_j)$ is both efficient and practical for inference. Unlike iteratively optimizing $\boldsymbol{\Lambda}$ and $\boldsymbol{\mu}, \mathbf{C}$ as in variational EM, [28] noticed that the calculation of the gradient w.r.t the lower bound indicates the updates of $\boldsymbol{\Lambda}$ can be analytically worked out by variational parameters, thus resulting a new objective function for the representation of lower bound that only relies on $\boldsymbol{\mu}, \mathbf{C}$ (details refer to [28]). We apply our algorithm to this variational logistic regression on three appropriate datasets: `DukeBreast` and `Leukemia` are small size but high-dimensional for sparse logistic regression, and `a9a` which is large. See Table 1 for additional dataset descriptions.

Fig. 1 shows the convergence of Gaussian variational lower bound for Bayesian logistic regression in terms of running time. It is worth mentioning that the lower bound of HFSGVI converges within 3 iterations on the small datasets `DukeBreast` and `Leukemia`. This is because all data points are fed to all algorithms and the HFSGVI uses a better approximation of the Hessian matrix to proceed $2^{\text{nd}}$ order optimization. L-BFGS-SGVI also take less time to converge and yield slightly larger lower bound than DSVI. In addition, as an SGD-based algorithm, it is clearly seen that DSVI is less stable for small datasets and fluctuates strongly even at the later optimized stage. For the large `a9a`, we observe that HFSGVI also needs 1000 iterations to reach a good lower bound and becomes less stable than the other two algorithms. However, L-BFGS-SGVI performs the best

Table 1: Comparison on number of misclassification

| Dataset(size: #train/test/feature) | DSVI | | L-BFGS-SGVI | | HFSGVI | |
|---|---|---|---|---|---|---|
| | train | test | train | test | train | test |
| DukeBreast(38/4/7129) | 0 | 2 | 0 | 1 | 0 | 0 |
| Leukemia(38/34/7129) | 0 | 3 | 0 | 3 | 0 | 3 |
| A9a(32561/16281/123) | 4948 | 2455 | 4936 | 2427 | 4931 | 2468 |

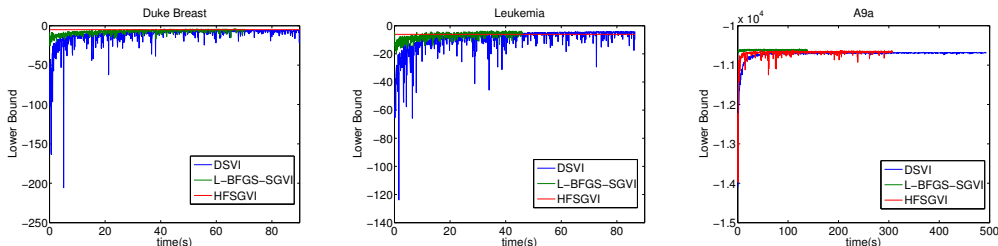

Figure 1: Convergence rate on logistic regression (zoom out or see larger figures in supplementary)

both in terms of convergence rate and the final lower bound. The misclassification report in Table 1 reflects the similar advantages of our approach, indicating a competitive predication ability on various datasets. Finally, it is worth mentioning that all three algorithms learn a set of very sparse regression coefficients on the three datasets (see supplement for additional visualizations).

## 4.2 Variational Auto-encoder

We also apply the 2nd order stochastic variational inference to train a VAE model (setting $M = 1$ for Monte Carlo integration to estimate expectation) or the equivalent deep neural networks with hybrid hidden layers. The datasets we used are images from the Frey Face, Olivetti Face and MNIST. We mainly learned three tasks by maximizing the variational lower bound: parameter estimation, images reconstruction and images generation. Meanwhile, we compared the convergence rate (running time) of three algorithms, where in this section the compared SGD is the Ada version [6] that is recommended for VAE model in [17, 25]. The experimental setting is as follows. The initial weights are randomly drawn from $\mathcal{N}(\mathbf{0}, 0.01^2\mathbf{I})$ or $\mathcal{N}(\mathbf{0}, 0.001^2\mathbf{I})$, while all bias terms are initialized as 0. The variational lower bound only introduces the regularization on the encoder parameters, so we add an $\mathcal{L}_2$ regularizer on decoder parameters with a shrinkage parameter 0.001 or 0.0001. The number of hidden nodes for encoder and decoder is the same for all auto-encoder model, which is reasonable and convenient to construct a symmetric structure. The number is always tuned from 200 to 800 with 100 increment. The mini-batch size is 100 for L-BFGS and Ada, while larger mini-batch is recommended for HF, meaning it should vary according to the training size.

The detailed results are shown in Fig. 2 and 3. Both Hessian-free and L-BFGS converge faster than Ada in terms of running time. HFSGVI also performs competitively with respet to generalization on testing data. Ada takes at least four times as long to achieve similar lower bound. Theoretically, Newton's method has a quadratic convergence rate in terms of iteration, but with a cubic algorithmic complexity at each iteration. However, we manage to lower the computation in each iteration to linear complexity. Thus considering the number of evaluated training data points, the 2nd order algorithm needs much fewer step than 1st order gradient descent (see visualization in supplementary on MNIST). The Hessian matrix also replaces manually tuned learning rates, and the affine invariant property allows for automatic learning rate adjustment. Technically, if the program can run in parallel with GPU, the speed advantages of 2nd order algorithm should be more obvious [21].

Fig. 2(b) and Fig. 3(b) are reconstruction results of input images. From the perspective of deep neural network, the only difference is the Gaussian distributed latent variables $\mathbf{z}$. By corollary of Theorem 2, we can roughly tell the mean $\boldsymbol{\mu}$ is able to represent the quantity of $\mathbf{z}$, meaning this layer is actually a linear transformation with noise, which looks like dropout training [5]. Specifically, Olivetti includes 64×64 pixels faces of various persons, which means more complicated models or preprocessing [13] (e.g. nearest neighbor interpolation, patch sampling) is needed. However, even when simply learning a very bottlenecked auto-encoder, our approach can achieve acceptable results. Note that although we have tuned the hyperparameters of Ada by cross-validation, the best result is still a bunch of mean faces. For manifold learning, Fig. 2(c) represents how the

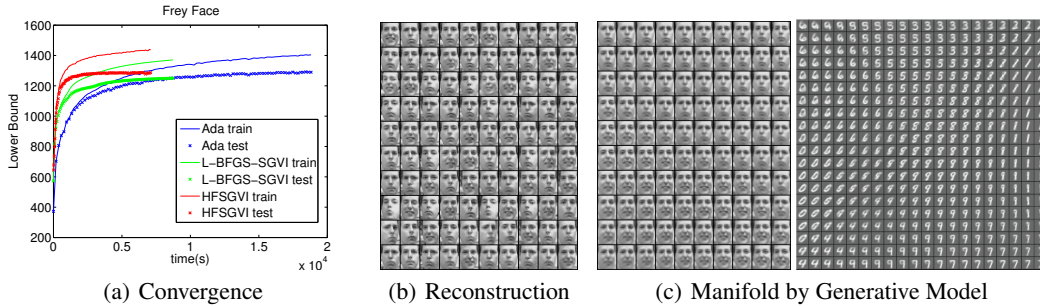

| (a) Convergence | (b) Reconstruction | (c) Manifold by Generative Model |

Figure 2: (a) shows how lower bound increases w.r.t program running time for different algorithms; (b) illustrates the reconstruction ability of this auto-encoder model when $d_z = 20$ (left 5 columns are randomly sampled from dataset); (c) is the learned manifold of generative model when $d_z = 2$.

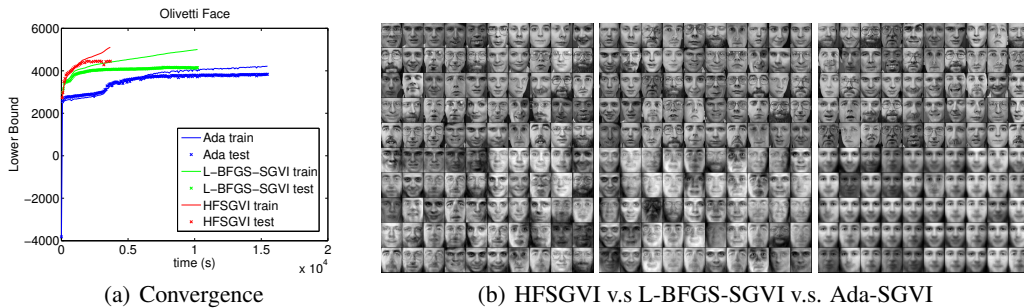

| (a) Convergence | (b) HFSGVI v.s L-BFGS-SGVI v.s. Ada-SGVI |

Figure 3: (a) shows running time comparison; (b) illustrates reconstruction comparison **without** patch sampling, where $d_z = 100$: top 5 rows are original faces.

learned generative model can simulate the images by HFSGVI. To visualize the results, we choose the 2D latent variable $\mathbf{z}$ in $p_{\boldsymbol{\psi}}(\mathbf{x}|\mathbf{z})$, where the parameter $\boldsymbol{\psi}$ is estimated by the algorithm. The two coordinates of $\mathbf{z}$ take values that were transformed through the inverse CDF of the Gaussian distribution from equal distance grid ($10\times10$ or $20\times20$) on the unit square. Then we merely use the generative model to simulate the images. Besides these learning tasks, denoising, imputation [25] and even generalizing to semi-supervised learning [16] are possible application of our approach.

# 5 Conclusions and Discussion

In this paper we proposed a scalable 2nd order stochastic variational method for generative models with continuous latent variables. By developing Gaussian backpropagation through reparametrization we introduced an efficient unbiased estimator for higher order gradients information. Combining with the efficient technique for computing Hessian-vector multiplication, we derived an efficient inference algorithm (HFSGVI) that allows for joint optimization of all parameters. The algorithmic complexity of each parameter update is quadratic w.r.t the dimension of latent variables for both 1st and 2nd derivatives. Furthermore, the overall computational complexity of our 2nd order SGVI is linear w.r.t the number of parameters in real applications just like SGD or Ada. However, HFSGVI may not behave as fast as Ada in some situations, e.g., when the pixel values of images are sparse due to fast matrix multiplication implementation in most softwares.

Future research will focus on some difficult deep models such as RNNs [10, 27] or Dynamic SBN [9]. Because of conditional independent structure by giving sampled latent variables, we may construct blocked Hessian matrix to optimize such dynamic models. Another possible area of future work would be reinforcement learning (RL) [20]. Many RL problems can be reduced to compute gradients of expectations (e.g., in policy gradient methods) and there has been series of exploration in this area for natural gradients. However, we would suggest that it might be interesting to consider where stochastic backpropagation fits in our framework and how 2nd order computations can help.

**Acknolwedgement** This research was supported in part by the Research Grants Council of the Hong Kong Special Administrative Region (Grant No. 614513).

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
