[Supplementary Material · supplementary_2ndOrder.pdf]

# Supplementary Material: Fast Second-Order Stochastic Backpropagation for Variational Inference

**Kai Fan**
Duke University
kai.fan@stat.duke.edu

**Ziteng Wang**[*]
Hong Kong University of Science and Technology
wangzt2012@gmail.com

**Jeffrey Beck**
Duke University
jeff.beck@duke.edu

**James Kwok**
Hong Kong University of Science and Technology
jamesk@cse.ust.hk

**Katherine Heller**
Duke University
kheller@gmail.com

## A  Proofs of the Extending Gaussian Gradient Equations

**Lemma A.1.** *Let $f(\mathbf{z}) : \mathcal{R}^{d_z} \to \mathcal{R}$ be an integrable and twice differentiable function. The second gradient of the expectation of $f(\mathbf{z})$ under a Gaussian distribution $\mathcal{N}(\mathbf{z}|\boldsymbol{\mu}, \mathbf{C})$ with respect to the mean $\boldsymbol{\mu}$ can be expressed as the expectation of the Hessian of $f(\mathbf{z})$:*

$$\nabla^2_{\mu_i,\mu_j} \mathbb{E}_{\mathcal{N}(\mathbf{z}|\boldsymbol{\mu},\mathbf{C})}[f(\mathbf{z})] = \mathbb{E}_{\mathcal{N}(\mathbf{z}|\boldsymbol{\mu},\mathbf{C})}[\nabla^2_{z_i,z_j} f(\mathbf{z})] = 2\nabla_{C_{ij}} \mathbb{E}_{\mathcal{N}(\mathbf{z}|\boldsymbol{\mu},\mathbf{C})}[f(\mathbf{z})]. \tag{1}$$

*Proof.* From Bonnet's theorem [1], we have

$$\nabla_{\mu_i} \mathbb{E}_{\mathcal{N}(\mathbf{z}|\boldsymbol{\mu},\mathbf{C})}[f(\mathbf{z})] = \mathbb{E}_{\mathcal{N}(\mathbf{z}|\boldsymbol{\mu},\mathbf{C})}[\nabla_{z_i} f(\mathbf{z})]. \tag{2}$$

Moreover, we can get the second order derivative,

$$\begin{aligned}
\nabla^2_{\mu_i,\mu_j} \mathbb{E}_{\mathcal{N}(\mathbf{z}|\boldsymbol{\mu},\mathbf{C})}[f(\mathbf{z})] &= \nabla_{\mu_i} \left( \mathbb{E}_{\mathcal{N}(\mathbf{z}|\boldsymbol{\mu},\mathbf{C})}[\nabla_{z_j} f(\mathbf{z})] \right) \\
&= \int \nabla_{\mu_i} \mathcal{N}(\mathbf{z}|\boldsymbol{\mu},\mathbf{C}) \nabla_{z_j} f(\mathbf{z}) \mathrm{d}\mathbf{z} \\
&= -\int \nabla_{z_i} \mathcal{N}(\mathbf{z}|\boldsymbol{\mu},\mathbf{C}) \nabla_{z_j} f(\mathbf{z}) \mathrm{d}\mathbf{z} \\
&= -\left[ \int \mathcal{N}(\mathbf{z}|\boldsymbol{\mu},\mathbf{C}) \nabla_{z_j} f(\mathbf{z}) \mathrm{d}z_{\neg i} \right]^{z_i=+\infty}_{z_i=-\infty} + \int \mathcal{N}(\mathbf{z}|\boldsymbol{\mu},\mathbf{C}) \nabla_{z_i,z_j} f(\mathbf{z}) \mathrm{d}\mathbf{z} \\
&= \mathbb{E}_{\mathcal{N}(\mathbf{z}|\boldsymbol{\mu},\mathbf{C})}[\nabla^2_{z_i,z_j} f(\mathbf{z})] \\
&= 2\nabla_{C_{ij}} \mathbb{E}_{\mathcal{N}(\mathbf{z}|\boldsymbol{\mu},\mathbf{C})}[f(\mathbf{z})],
\end{aligned}$$

where the last euality we use the equation

$$\nabla_{C_{ij}} \mathcal{N}(\mathbf{z}|\boldsymbol{\mu},\mathbf{C}) = \frac{1}{2} \nabla^2_{z_i,z_j} \mathcal{N}(\mathbf{z}|\boldsymbol{\mu},\mathbf{C}). \tag{3}$$

□

---

[*]Equal Contribution to this work.

**Lemma A.2.** *Let $f(\mathbf{z}) : \mathcal{R}^{d_z} \to \mathcal{R}$ be an integrable and fourth differentiable function. The second gradient of the expectation of $f(\mathbf{z})$ under a Gaussian distribution $\mathcal{N}(\mathbf{z}|\boldsymbol{\mu}, \mathbf{C})$ with respect to the covariance $\mathbf{C}$ can be expressed as the expectation of the forth gradient of $f(\mathbf{z})$*

$$\nabla^2_{C_{i,j}, C_{k,l}} \mathbb{E}_{\mathcal{N}(\mathbf{z}|\boldsymbol{\mu}, \mathbf{C})}[f(\mathbf{z})] = \frac{1}{4} \mathbb{E}_{\mathcal{N}(\mathbf{z}|\boldsymbol{\mu}, \mathbf{C})}[\nabla^4_{z_i, z_j, z_k, z_l} f(\mathbf{z})]. \tag{4}$$

*Proof.* From Price's theorem [3], we have

$$\nabla_{C_{i,j}} \mathbb{E}_{\mathcal{N}(\mathbf{z}|\boldsymbol{\mu}, \mathbf{C})}[f(\mathbf{z})] = \frac{1}{2} \mathbb{E}_{\mathcal{N}(\mathbf{z}|\boldsymbol{\mu}, \mathbf{C})}[\nabla^2_{z_i, z_j} f(\mathbf{z})]. \tag{5}$$

$$
\begin{aligned}
\nabla^2_{C_{i,j}, C_{k,l}} \mathbb{E}_{\mathcal{N}(\mathbf{z}|\boldsymbol{\mu}, \mathbf{C})}[f(\mathbf{z})] &= \frac{1}{2} \nabla_{C_{k,l}} \left( \mathbb{E}_{\mathcal{N}(\mathbf{z}|\boldsymbol{\mu}, \mathbf{C})}[\nabla^2_{z_i, z_j} f(\mathbf{z})] \right) \\
&= \frac{1}{2} \int \nabla_{C_{k,l}} \mathcal{N}(\mathbf{z}|\boldsymbol{\mu}, \mathbf{C}) \nabla^2_{z_i, z_j} f(\mathbf{z}) \mathrm{d}\mathbf{z} \\
&= \frac{1}{4} \int \nabla^2_{z_k, z_l} \mathcal{N}(\mathbf{z}|\boldsymbol{\mu}, \mathbf{C}) \nabla^2_{z_i, z_j} f(\mathbf{z}) \mathrm{d}\mathbf{z} \\
&= \frac{1}{4} \int \mathcal{N}(\mathbf{z}|\boldsymbol{\mu}, \mathbf{C}) \nabla^4_{z_i, z_j, z_k, z_l} f(\mathbf{z}) \mathrm{d}\mathbf{z} \\
&= \frac{1}{4} \mathbb{E}_{\mathcal{N}(\mathbf{z}|\boldsymbol{\mu}, \mathbf{C})}[\nabla^4_{z_i, z_j, z_k, z_l} f(\mathbf{z})].
\end{aligned}
$$

In the third equality we use the Eq.(3) again. For the fourth equality we use the product rule for integrals twice. $\qquad\square$

From Eq.(2) and Eq.(5) we can straightforward write the second order gradient of interaction term as well:

$$\nabla^2_{\mu_i, C_{k,l}} \mathbb{E}_{\mathcal{N}(\mu, \mathbf{C})}[f(z)] = \frac{1}{2} \mathbb{E}_{\mathcal{N}(\mu, \mathbf{C})} \left[ \nabla^3_{z_i, z_k, z_l} f(z) \right]. \tag{6}$$

## B   Proof of Theorem 1

By using the linear transformation $\mathbf{z} = \boldsymbol{\mu} + \mathbf{R}\boldsymbol{\epsilon}$, where $\boldsymbol{\epsilon} \sim N(0, \mathbf{I}_{d_z})$, we can generate samples form any Gaussian distribution $\mathcal{N}(\boldsymbol{\mu}, \mathbf{C})$, $\mathbf{C} = \mathbf{R}\mathbf{R}^\top$, where $\boldsymbol{\mu}(\boldsymbol{\theta}), \mathbf{R}(\boldsymbol{\theta})$ are both dependent on parameter $\boldsymbol{\theta} = (\theta_l)_{l=1}^d$.

Then the gradients of the expectation with respect to $\boldsymbol{\mu}$ and (or) $\mathbf{R}$ is

$$
\begin{aligned}
\nabla_{\mathbf{R}} \mathbb{E}_{\mathcal{N}(\boldsymbol{\mu}, \mathbf{C})}[f(\mathbf{z})] &= \nabla_{\mathbf{R}} \mathbb{E}_{\mathcal{N}(0, \mathbf{I})}[f(\boldsymbol{\mu} + \mathbf{R}\boldsymbol{\epsilon})] = \mathbb{E}_{\mathcal{N}(0, \mathbf{I})}[\boldsymbol{\epsilon} \mathbf{g}^\top] \\
\nabla^2_{R_{i,j}, R_{k,l}} \mathbb{E}_{\mathcal{N}(\boldsymbol{\mu}, \mathbf{C})}[f(\mathbf{z})] &= \nabla_{R_{i,j}} \mathbb{E}_{\mathcal{N}(0, \mathbf{I})}[\epsilon_l g_k] = \mathbb{E}_{\mathcal{N}(0, \mathbf{I})}[\epsilon_j \epsilon_l H_{ik}] \\
\nabla^2_{\mu_i, R_{k,l}} \mathbb{E}_{\mathcal{N}(\boldsymbol{\mu}, \mathbf{C})}[f(\mathbf{z})] &= \nabla_{\mu_i} \mathbb{E}_{\mathcal{N}(0, \mathbf{I})}[\epsilon_l g_k] = \mathbb{E}_{\mathcal{N}(0, \mathbf{I})}[\epsilon_l H_{ik}] \\
\nabla^2_{\boldsymbol{\mu}} \mathbb{E}_{\mathcal{N}(\boldsymbol{\mu}, \mathbf{C})}[f(\mathbf{z})] &= \mathbb{E}_{\mathcal{N}(0, \mathbf{I})}[\mathbf{H}]
\end{aligned}
$$

where $\mathbf{g} = \{g_j\}_{j=1}^{d_z}$ is the gradient of $f$ evaluated at $\boldsymbol{\mu} + \mathbf{R}\boldsymbol{\epsilon}$, $\mathbf{H} = \{H_{ij}\}_{d_z \times d_z}$ is the Hessian of $f$ evaluated at $\boldsymbol{\mu} + \mathbf{R}\boldsymbol{\epsilon}$.

Furthermore, we write the second order derivatives into matrix form:

$$
\begin{aligned}
\nabla^2_{\boldsymbol{\mu}, \mathbf{R}} \mathbb{E}_{\mathcal{N}(\boldsymbol{\mu}, \mathbf{C})}[f(\mathbf{z})] &= \mathbb{E}_{\mathcal{N}(0, \mathbf{I})}[\boldsymbol{\epsilon}^\top \otimes \mathbf{H}], \\
\nabla^2_{\mathbf{R}} \mathbb{E}_{\mathcal{N}(\boldsymbol{\mu}, \mathbf{C})}[f(\mathbf{z})] &= \mathbb{E}_{\mathcal{N}(0, \mathbf{I})}[(\boldsymbol{\epsilon}\boldsymbol{\epsilon}^T) \otimes \mathbf{H}].
\end{aligned}
$$

For a particular model, such as deep generative model, $\boldsymbol{\mu}$ and $\mathbf{C}$ are depend on the model parameters, we denote them as $\boldsymbol{\theta} = (\theta_l)_{l=1}^d$, i.e. $\boldsymbol{\mu} = \boldsymbol{\mu}(\boldsymbol{\theta}), \mathbf{C} = \mathbf{C}(\boldsymbol{\theta})$. Combining Eq.2 and Eq.5 and using the chain rule we have

$$\nabla_{\theta_l} \mathbb{E}_{\mathcal{N}(\boldsymbol{\mu}, \mathbf{C})}[f(\mathbf{z})] = \mathbb{E}_{\mathcal{N}(\boldsymbol{\mu}, \mathbf{C})} \left[ \mathbf{g}^\top \frac{\partial \boldsymbol{\mu}}{\partial \theta_l} + \frac{1}{2} \operatorname{Tr} \left( \mathbf{H} \frac{\partial \mathbf{C}}{\partial \theta_l} \right) \right],$$

where $\mathbf{g}$ and $\mathbf{H}$ are the first and second order gradient of $f(\mathbf{z})$ for abusing notation. This formulation involves matrix-matrix product, resulting in an algorithmic complexity $\mathcal{O}(d_z^2)$ for any single element of $\boldsymbol{\theta}$ w.r.t $f(\mathbf{z})$, and $\mathcal{O}(dd_z^2)$, $\mathcal{O}(d^2 d_z^2)$ for overall gradient and Hessian respectively.

Considering $\mathbf{C} = \mathbf{R}\mathbf{R}^\top$, $\mathbf{z} = \boldsymbol{\mu} + \mathbf{R}\boldsymbol{\epsilon}$,

$$
\begin{aligned}
\nabla_{\theta_l} \mathbb{E}_{\mathcal{N}(\boldsymbol{\mu},\mathbf{C})}[f(\mathbf{z})] &= \mathbb{E}_{\mathcal{N}(0,\mathbf{I})}\left[\mathbf{g}^\top \frac{\partial \boldsymbol{\mu}}{\partial \theta_l} + \mathrm{Tr}\left(\boldsymbol{\epsilon}\mathbf{g}^\top \frac{\partial \mathbf{R}}{\partial \theta_l}\right)\right] \\
&= \mathbb{E}_{\mathcal{N}(0,\mathbf{I})}\left[\mathbf{g}^\top \frac{\partial \boldsymbol{\mu}}{\partial \theta_l} + \mathbf{g}^\top \frac{\partial \mathbf{R}}{\partial \theta_l}\boldsymbol{\epsilon}\right].
\end{aligned}
$$

For the second order, we have the following separated formulation:

$$
\begin{aligned}
\nabla^2_{\theta_{l_1}\theta_{l_2}} \mathbb{E}_{\mathcal{N}(\boldsymbol{\mu},\mathbf{C})}[f(\mathbf{z})] &= \nabla_{\theta_{l_1}} \mathbb{E}_{\mathcal{N}(0,\mathbf{I})}\left[\sum_i g_i \frac{\partial \mu_i}{\partial \theta_{l_2}} + \sum_{i,j} \epsilon_j g_i \frac{\partial R_{ij}}{\partial \theta_{l_2}}\right] \\
&= \mathbb{E}_{\mathcal{N}(0,\mathbf{I})}\left[\sum_{i,j} H_{ji}\left(\frac{\partial \mu_j}{\partial \theta_{l_1}} + \sum_k \epsilon_k \frac{\partial R_{jk}}{\partial \theta_{l_1}}\right)\frac{\partial \mu_i}{\partial \theta_{l_2}} + \sum_i g_i \frac{\partial^2 \mu_i}{\partial \theta_{l_1}\partial \theta_{l_2}}\right. \\
&\quad + \left.\sum_{i,j}\epsilon_j\left(\sum_k H_{ik}\left(\frac{\partial \mu_k}{\partial \theta_{l_1}} + \sum_l \epsilon_l \frac{\partial R_{kl}}{\partial \theta_{l_1}}\right)\right)\frac{\partial R_{ij}}{\partial \theta_{l_2}} + \sum_{i,j}\epsilon_j g_i \frac{\partial^2 R_{ij}}{\partial \theta_{l_1}\partial \theta_{l_2}}\right] \\
&= \mathbb{E}_{\mathcal{N}(0,\mathbf{I})}\left[\frac{\partial \boldsymbol{\mu}}{\partial \theta_{l_1}}^\top \mathbf{H}\frac{\partial \boldsymbol{\mu}}{\partial \theta_{l_2}} + \left(\frac{\partial \mathbf{R}}{\partial \theta_{l_1}}\boldsymbol{\epsilon}\right)^\top \mathbf{H}\frac{\partial \boldsymbol{\mu}}{\partial \theta_{l_2}} + \mathbf{g}^\top \frac{\partial^2 \boldsymbol{\mu}}{\partial \theta_{l_1}\partial \theta_{l_2}}\right. \\
&\quad + \left.\left(\frac{\partial \mathbf{R}}{\partial \theta_{l_2}}\boldsymbol{\epsilon}\right)^\top \mathbf{H}\frac{\partial \boldsymbol{\mu}}{\partial \theta_{l_1}} + \left(\frac{\partial \mathbf{R}}{\partial \theta_{l_1}}\boldsymbol{\epsilon}\right)^\top \mathbf{H}\frac{\partial \mathbf{R}}{\partial \theta_{l_2}}\boldsymbol{\epsilon} + \mathbf{g}^\top \frac{\partial^2 \mathbf{R}}{\partial \theta_{l_1}\partial \theta_{l_2}}\boldsymbol{\epsilon}\right] \\
&= \mathbb{E}_{\mathcal{N}(0,\mathbf{I})}\left[\frac{\partial(\boldsymbol{\mu}+\mathbf{R}\boldsymbol{\epsilon})}{\partial \theta_{l_1}}^\top \mathbf{H}\frac{\partial(\boldsymbol{\mu}+\mathbf{R}\boldsymbol{\epsilon})}{\partial \theta_{l_2}} + \mathbf{g}^\top \frac{\partial^2(\boldsymbol{\mu}+\mathbf{R}\boldsymbol{\epsilon})}{\partial \theta_{l_1}\partial \theta_{l_2}}\right].
\end{aligned}
$$

It is noticed that for second order gradient computation, it only involves matrix-vector or vector-vector multiplication, thus leading to an algorithmic complexity $\mathcal{O}(d_z^2)$ for each pair of $\boldsymbol{\theta}$.

One practical parametrization is $\mathbf{C} = \mathrm{diag}\{\sigma_1^2, ..., \sigma_{d_z}^2\}$ or $\mathbf{R} = \mathrm{diag}\{\sigma_1, ..., \sigma_{d_z}\}$, which will reduce the actual second order gradient computation complexity, albeit the same order of $\mathcal{O}(d_z^2)$. Then we have

$$
\begin{aligned}
\nabla_{\theta_l} \mathbb{E}_{\mathcal{N}(\boldsymbol{\mu},\mathbf{C})}[f(\mathbf{z})] &= \mathbb{E}_{\mathcal{N}(0,\mathbf{I})}\left[\mathbf{g}^\top \frac{\partial \boldsymbol{\mu}}{\partial \theta_l} + \sum_i \epsilon_i g_i \frac{\partial \sigma_i}{\partial \theta_l}\right] \\
&= \mathbb{E}_{\mathcal{N}(0,\mathbf{I})}\left[\mathbf{g}^\top \frac{\partial \boldsymbol{\mu}}{\partial \theta_l} + (\boldsymbol{\epsilon}\odot\mathbf{g})^\top \frac{\partial \boldsymbol{\sigma}}{\partial \theta_l}\right], \quad\quad (7) \\
\nabla^2_{\theta_{l_1}\theta_{l_2}} \mathbb{E}_{\mathcal{N}(\boldsymbol{\mu},\mathbf{C})}[f(\mathbf{z})] &= \mathbb{E}_{\mathcal{N}(0,\mathbf{I})}\left[\frac{\partial \boldsymbol{\mu}}{\partial \theta_{l_1}}^\top \mathbf{H}\frac{\partial \boldsymbol{\mu}}{\partial \theta_{l_2}} + \left(\boldsymbol{\epsilon}\odot\frac{\partial \boldsymbol{\sigma}}{\partial \theta_{l_1}}\right)^\top \mathbf{H}\frac{\partial \boldsymbol{\mu}}{\partial \theta_{l_2}} + \mathbf{g}^\top \frac{\partial^2 \boldsymbol{\mu}}{\partial \theta_{l_1}\partial \theta_{l_2}}\right. \\
&\quad + \left(\boldsymbol{\epsilon}\odot\frac{\partial \boldsymbol{\sigma}}{\partial \theta_{l_2}}\right)^\top \mathbf{H}\frac{\partial \boldsymbol{\mu}}{\partial \theta_{l_1}} + \left(\boldsymbol{\epsilon}\odot\frac{\partial \boldsymbol{\sigma}}{\partial \theta_{l_1}}\right)^\top \mathbf{H}\left(\boldsymbol{\epsilon}\odot\frac{\partial \boldsymbol{\sigma}}{\partial \theta_{l_2}}\right) \\
&\quad + \left.(\boldsymbol{\epsilon}\odot\mathbf{g})^\top \frac{\partial^2 \boldsymbol{\sigma}}{\partial \theta_{l_1}\partial \theta_{l_2}}\right] \\
&= \mathbb{E}_{\mathcal{N}(0,\mathbf{I})}\left[\left(\frac{\partial \boldsymbol{\mu}}{\partial \theta_{l_1}} + \boldsymbol{\epsilon}\odot\frac{\partial \boldsymbol{\sigma}}{\partial \theta_{l_1}}\right)^\top \mathbf{H}\left(\frac{\partial \boldsymbol{\mu}}{\partial \theta_{l_2}} + \boldsymbol{\epsilon}\odot\frac{\partial \boldsymbol{\sigma}}{\partial \theta_{l_2}}\right)\right. \\
&\quad + \left.\mathbf{g}^\top\left(\frac{\partial^2 \boldsymbol{\mu}}{\partial \theta_{l_1}\partial \theta_{l_2}} + \frac{\partial^2(\boldsymbol{\epsilon}\odot\boldsymbol{\sigma})}{\partial \theta_{l_1}\partial \theta_{l_2}}\right)\right], \quad\quad (8)
\end{aligned}
$$

where $\odot$ is Hadamard (or element-wise) product, and $\boldsymbol{\sigma} = (\sigma_1, ..., \sigma_{d_z})^\top$.

**Derivation for Hessian-Free SGVI without $\theta$ Plugging** This means $(\boldsymbol{\mu}, \mathbf{R})$ is the parameter for variational distribution. According the derivation in this section, the Hessian matrix with respect to $(\boldsymbol{\mu}, \mathbf{R})$ can represented as $\mathbf{H}_{\boldsymbol{\mu}, \mathbf{R}} = \mathbb{E}_{\mathcal{N}(0, \mathbf{I})} \left[ \left( \begin{bmatrix} 1 \\ \boldsymbol{\epsilon} \end{bmatrix} [1, \boldsymbol{\epsilon}^\top] \right) \otimes \mathbf{H} \right]$. For any $d_z \times (d_z + 1)$ matrix $\mathbf{V}$ with the same dimensionality of $[\boldsymbol{\mu}, \mathbf{R}]$, we also have the Hessian-vector multiplication equation.

$$\mathbf{H}_{\boldsymbol{\mu}, \mathbf{R}} vec(\mathbf{V}) = \mathbb{E}_{\mathcal{N}(0, \mathbf{I})} \left[ vec \left( \mathbf{H} \mathbf{V} \begin{bmatrix} 1 \\ \boldsymbol{\epsilon} \end{bmatrix} [1, \boldsymbol{\epsilon}^\top] \right) \right]$$

where $vec(\cdot)$ denotes the vectorization of the matrix formed by stacking the columns into a single column vector. This allows an efficient computation both in speed and storage.

# C    Forward-Backward Algorithm for Special Variation Auto-encoder Model

We illustrate the equivalent deep neural network model (Figure 1) by setting $M = 1$ in VAE, and derive the gradient computation by lawyer-wise backpropagation. Without generalization, we give discussion on the binary input and diagonal covariance matrix, while it is straightforward to write the continuous case. For binary input, the parameters are $\{(W_i, b_i)\}_{i=1}^5$.

The feedforward process is as follows:

$$
\begin{aligned}
\mathbf{h}_e &= \tanh(W_1 \mathbf{x} + b_1) \\
\boldsymbol{\mu}_e &= W_2 \mathbf{h}_e + b_2 \\
\log \boldsymbol{\sigma}_e &= 0.5 * (W_3 \mathbf{h}_e + b_3) \\
\boldsymbol{\epsilon} &\sim \mathcal{N}(0, \mathbf{I}_{d_z}) \\
\mathbf{z} &= \boldsymbol{\mu}_e + \boldsymbol{\sigma}_e \odot \boldsymbol{\epsilon} \\
\mathbf{h}_d &= \tanh(W_4 \mathbf{z} + b_4) \\
\mathbf{y} &= \text{sigmoid}(W_5 \mathbf{h}_d + b_5).
\end{aligned}
$$

Considering the cross-entropy loss function, the backward process for gradient backpropagation computation is:

$$
\begin{aligned}
\boldsymbol{\delta}_5 &= \mathbf{x} \odot (1 - \mathbf{y}) + (1 - \mathbf{x}) \odot \mathbf{y} \\
\nabla_{W_5} &= \boldsymbol{\delta}_5 \mathbf{h}_d^\top, \quad \nabla_{b_5} = \boldsymbol{\delta}_5 \\
\boldsymbol{\delta}_4 &= (W_5^\top \boldsymbol{\delta}_5) \odot (\mathbf{1} - \mathbf{h}_d \odot \mathbf{h}_d) \\
\nabla_{W_4} &= \boldsymbol{\delta}_4 \mathbf{z}^\top, \quad \nabla_{b_4} = \boldsymbol{\delta}_4 \\
\boldsymbol{\delta}_3 &= 0.5 * [(W_4^\top \boldsymbol{\delta}_4) \odot (\mathbf{z} - \boldsymbol{\mu}_e) + \mathbf{1} - \boldsymbol{\sigma}_e^2] \\
\nabla_{W_3} &= \boldsymbol{\delta}_3 \mathbf{h}_e^\top, \quad \nabla_{b_3} = \boldsymbol{\delta}_3 \\
\boldsymbol{\delta}_2 &= W_4^\top \boldsymbol{\delta}_4 - \boldsymbol{\mu}_e \\
\nabla_{W_2} &= \boldsymbol{\delta}_2 \mathbf{h}_e^\top, \quad \nabla_{b_2} = \boldsymbol{\delta}_2 \\
\boldsymbol{\delta}_1 &= (W_2^\top \boldsymbol{\delta}_2 + W_3^\top \boldsymbol{\delta}_3) \odot (\mathbf{1} - \mathbf{h}_e \odot \mathbf{h}_e) \\
\nabla_{W_1} &= \boldsymbol{\delta}_1 \mathbf{x}^\top, \quad \nabla_{b_1} = \boldsymbol{\delta}_1.
\end{aligned}
$$

Notice that when we compute the differences $\boldsymbol{\delta}_2, \boldsymbol{\delta}_3$, we also include the prior term which acts as the role of regularization penalty. In addition, we can add the $\mathcal{L}_2$ penalty to the weight matrix as well. The only modification is to change the expression of $\nabla_{W_i}$ by adding $\lambda W_i$, where $\lambda$ is a tunable hyper-parameter.

# D    Variance Analysis (Proof of Theorem 2)

In this part we analyze the variance of the stochastic estimator.

Figure 1: Auto-encoder Model by Deep Neural Nets.

**Lemma D.1.** *For any convex function $\phi$,*

$$\mathbb{E}[\phi(f(\boldsymbol{\epsilon}) - \mathbb{E}[f(\boldsymbol{\epsilon})])] \leq \mathbb{E}\left[\phi\left(\frac{\pi}{2}\langle\nabla f(\boldsymbol{\epsilon}), \boldsymbol{\eta}\rangle\right)\right], \tag{9}$$

*where $\boldsymbol{\epsilon}, \boldsymbol{\eta} \sim \mathcal{N}(0, \mathbf{I}_{d_z})$ and $\boldsymbol{\epsilon}, \boldsymbol{\eta}$ are independent.*

*Proof.* Using interpolation $\boldsymbol{\gamma}(\omega) = \boldsymbol{\epsilon}\sin(\omega) + \boldsymbol{\eta}\cos(\omega)$, then $\boldsymbol{\gamma}'(\omega) = \boldsymbol{\epsilon}\cos(\omega) - \boldsymbol{\eta}\sin(\omega)$, and $\boldsymbol{\gamma}(0) = \boldsymbol{\eta}, \boldsymbol{\gamma}(\pi/2) = \boldsymbol{\epsilon}$. Furthermore, we have the equation,

$$f(\boldsymbol{\epsilon}) - f(\boldsymbol{\eta}) = \int_0^{\frac{\pi}{2}} \frac{\mathrm{d}}{\mathrm{d}\omega} f(\boldsymbol{\gamma}(\omega))\mathrm{d}\omega = \int_0^{\frac{\pi}{2}} \langle\nabla f(\boldsymbol{\gamma}(\omega)), \boldsymbol{\gamma}'(\omega)\rangle\mathrm{d}\omega.$$

Then

$$
\begin{aligned}
\mathbb{E}_{\boldsymbol{\epsilon}}[\phi(f(\boldsymbol{\epsilon}) - \mathbb{E}[f(\boldsymbol{\epsilon})])] &= \mathbb{E}_{\boldsymbol{\epsilon}}[\phi(f(\boldsymbol{\epsilon}) - \mathbb{E}_{\boldsymbol{\eta}}[f(\boldsymbol{\eta})])] \leq \mathbb{E}_{\boldsymbol{\epsilon},\boldsymbol{\eta}}[\phi(f(\boldsymbol{\epsilon}) - f(\boldsymbol{\eta}))] \\
&= \mathbb{E}\left[\phi\left(\frac{2}{\pi}\int_0^{\frac{\pi}{2}}\frac{\pi}{2}\langle\nabla f(\boldsymbol{\gamma}(\omega)), \boldsymbol{\gamma}'(\omega)\rangle\mathrm{d}\omega\right)\right] \\
&\leq \frac{2}{\pi}\mathbb{E}\left[\int_0^{\frac{\pi}{2}}\phi\left(\frac{\pi}{2}\langle\nabla f(\boldsymbol{\gamma}(\omega)), \boldsymbol{\gamma}'(\omega)\rangle\right)\mathrm{d}\omega\right] \\
&= \frac{2}{\pi}\int_0^{\frac{\pi}{2}}\mathbb{E}\left[\phi\left(\frac{\pi}{2}\langle\nabla f(\boldsymbol{\gamma}(\omega)), \boldsymbol{\gamma}'(\omega)\rangle\right)\right]\mathrm{d}\omega \\
&= \mathbb{E}\left[\phi\left(\frac{\pi}{2}\langle\nabla f(\boldsymbol{\epsilon}), \boldsymbol{\eta}\rangle\right)\right].
\end{aligned}
$$

The above two inequalities use the Jensen's Inequality. The last equation holds because both $\boldsymbol{\gamma}$ and $\boldsymbol{\gamma}'$ follow $\mathcal{N}(0, \mathbf{I}_d)$, and $\mathbb{E}[\boldsymbol{\gamma}\boldsymbol{\gamma}'^{\top}] = 0$ implies they are independent. $\qquad\square$

Before giving a dimensional free bound, we first let $\phi(x) = x^2$ and can obtain a relatively loosen bound of variance for our estimators. Assuming $f$ is a $L$-Lipschitz differentiable function and $\epsilon \sim \mathcal{N}(0, \mathbf{I}_{d_z})$, the following inequality holds:

$$\mathbb{E}[(f(\boldsymbol{\epsilon}) - \mathbb{E}[f(\boldsymbol{\epsilon})])^2] \leq \frac{\pi^2 L^2 d_z}{4}. \tag{10}$$

To see the reason, we only need to reuse the double sample trick and the expectation of Chi-squared distribution, we have

$$\mathbb{E}\left[\left(\frac{\pi}{2}\langle\nabla f(\boldsymbol{\epsilon}), \boldsymbol{\eta}\rangle\right)^2\right] \leq \frac{\pi^2 L^2}{4}\mathbb{E}[\|\boldsymbol{\eta}\|^2] = \frac{\pi^2 L^2 d_z}{4}.$$

Then by Lemma D.1, Eq.(11) holds. To get a tighter bound as in Theorem 2, we give the following Lemma D.2 and Lemma D.3 first.

**Lemma D.2** ([2]). *A random variable $X$ with mean $\mu = \mathbb{E}[X]$ is sub-Gaussian if there exists a positive number $\sigma$ such that for all $\lambda \in \mathcal{R}^+$*

$$\mathbb{E}\left[e^{\lambda(X-\mu)}\right] \leq e^{\sigma^2\lambda^2/2},$$

*then we have*

$$\mathbb{E}\left[(X-\mu)^2\right] \leq \sigma^2.$$

*Proof.* By Taylor's expansion,

$$\mathbb{E}\left[e^{\lambda(X-\mu)}\right] = \mathbb{E}\left[\sum_{i=1}^{\infty}\frac{\lambda^i}{i!}(X-\mu)^i\right] \leq e^{\sigma^2\lambda^2/2} = \sum_{i=0}^{\infty}\frac{\sigma^{2i}\lambda^{2i}}{2^i i!}.$$

Thus $\frac{\lambda^2}{2}\mathbb{E}[(X-\mu)^2] \leq \frac{\sigma^2\lambda^2}{2} + o(\lambda^2)$. Let $\lambda \to 0$, we have $\mathrm{Var}(X) \leq \sigma^2$. $\qquad\square$

**Lemma D.3.** *If $f(x)$ is a $L$-lipschitz differentiable function and $\boldsymbol{\epsilon} \in \mathcal{N}(0, \mathbf{I}_{d_z})$ then the random variable $f(\boldsymbol{\epsilon}) - \mathbb{E}[f(\boldsymbol{\epsilon})]$ is sub-Gaussian with parameter $L$, i.e. for all $\lambda \in \mathcal{R}^+$*

$$\mathbb{E}\left[e^{\lambda(f(\boldsymbol{\epsilon})-\mathbb{E}[f(\boldsymbol{\epsilon})])}\right] \leq e^{L^2\lambda^2\pi^2/8}.$$

*Proof.* From Lemma D.1, we have

$$
\begin{aligned}
\mathbb{E}\left[e^{\lambda(f(\boldsymbol{\epsilon})-\mathbb{E}[f(\boldsymbol{\epsilon})])}\right] \leq & \mathbb{E}_{\boldsymbol{\epsilon},\boldsymbol{\eta}}\left[e^{\lambda\frac{\pi}{2}\langle\nabla f(\boldsymbol{\epsilon}),\boldsymbol{\eta}\rangle}\right] \\
=& \mathbb{E}_{\boldsymbol{\epsilon},\boldsymbol{\eta}}\left[e^{\lambda\frac{\pi}{2}\sum_{i=1}^{d_z}\left(\eta_i\frac{\partial}{\partial\epsilon_i}f(\boldsymbol{\epsilon})\right)}\right] = \mathbb{E}_{\boldsymbol{\epsilon}}\left[e^{\sum_{i=1}^{d_z}\frac{1}{2}\left(\lambda\frac{\pi}{2}\frac{\partial}{\partial\epsilon_i}f(\boldsymbol{\epsilon})\right)^2}\right] = \mathbb{E}_{\boldsymbol{\epsilon}}\left[e^{\frac{\lambda^2\pi^2}{8}\|\nabla f(\boldsymbol{\epsilon})\|^2}\right] \\
\leq & \exp\left(\frac{\lambda^2\pi^2 L^2}{8}\right).
\end{aligned}
$$

$$\square$$

**Proof of Theorem 2** Combining Lemma D.2 and Lemma D.3 we complete the proof of Theorem 2.

In addition, we can also obtain a tail bound,

$$\mathbb{P}_{\boldsymbol{\epsilon}\sim\mathcal{N}(0,\mathbf{I}_{d_z})}\left(|f(\boldsymbol{\epsilon})-\mathbb{E}[f(\boldsymbol{\epsilon})]| \geq t\right) \leq 2e^{-\frac{2t^2}{\pi^2 L^2}}. \tag{11}$$

For $\lambda > 0$, Let $\boldsymbol{\epsilon}_1, \ldots, \boldsymbol{\epsilon}_M$ be i.i.d random variables with distribution $\mathcal{N}(0, \mathbf{I}_{d_z})$,

$$
\begin{aligned}
\mathbb{P}\left(\frac{1}{M}\sum_{m=1}^{M}f(\boldsymbol{\epsilon}_m)-\mathbb{E}[f(\boldsymbol{\epsilon})] \geq t\right) &=& \mathbb{P}\left(\sum_{m=1}^{M}f(\boldsymbol{\epsilon}_m)-M\mathbb{E}[f(\boldsymbol{\epsilon})] \geq Mt\right) \\
&=& \mathbb{P}\left(e^{\lambda\left(\sum_{m=1}^{M}f(\boldsymbol{\epsilon}_m)-M\mathbb{E}[f(\boldsymbol{\epsilon})]\right)} \geq e^{\lambda Mt}\right) \\
&\leq& \mathbb{E}\left[e^{\lambda\left(\sum_{m=1}^{M}f(\boldsymbol{\epsilon}_m)-M\mathbb{E}[f(\boldsymbol{\epsilon})]\right)}\right]e^{-\lambda Mt} \\
&=& \left(\mathbb{E}\left[e^{\lambda(f(\boldsymbol{\epsilon}_m)-\mathbb{E}[f(\boldsymbol{\epsilon})])}\right]e^{-\lambda t}\right)^{M}.
\end{aligned}
$$

According to Lemma D.3, let $\lambda = \frac{4t}{\pi^2 L^2}$, we have $\mathbb{P}\left(\frac{1}{M}\sum_{m=1}^{M}f(\boldsymbol{\epsilon}_m)-\mathbb{E}[f(\boldsymbol{\epsilon})] \geq t\right) \leq e^{-\frac{2Mt^2}{\pi^2 L^2}}$.
The other side can apply the same trick. Let $M = 1$ we have Inequality (11). Thus Theorem 2 and Inequality (11) provide the theoretical guarantee for stochastic method for Gaussian variables.

# E  More on Conjugate Gradient Descent

The preconditioned CG is used and theoretically the quantitative relation between the iteration $K$ and relative tolerance $e$ is $e < \exp(-2K/\sqrt{c})$ [4], where $c$ is matrix conditioner. Also the inequality indicates that the conditioner $c$ can be nearly as large as $O(K^2)$.

# F  Proof of Lemma 4

*Proof.* Since $g(x) = \frac{1}{1+e^{-x}}$, we have $g'(x) = g(x)(1 - g(x)) \leq \frac{1}{4}$.

$$|f(\boldsymbol{\epsilon}) - f(\boldsymbol{\eta})| = |g(h_i(\boldsymbol{\epsilon})) - g(h_i(\boldsymbol{\eta}))| \leq \frac{1}{4}|h_i(\boldsymbol{\epsilon}) - h_i(\boldsymbol{\eta})| \leq \frac{1}{4}\|W_{i,}\mathbf{R}\|_2\|\boldsymbol{\epsilon} - \boldsymbol{\eta}\|_2.$$

Since $\tanh(x) = 2g(2x) - 1$ and $\log(1 + e^x)' \leq 1$, the bound is trivial. □

# G  Experiments

All the experiments are conducted on a 3.2GHz CPU computer with X-Intel 32G RAM. For fair comparison, the algorithms and datasets we referred to as the baseline remain the same as in the previously cited work and software was downloaded from the website of relevant papers.

Datasets `DukeBreast`, `Leukemia` and `A9a` are downloaded from `http://www.csie.ntu.edu.tw/~cjlin/libsvmtools/datasets/binary.html`. Datasets Frey Face, Olivetti Face and MNIST are downloaded from `http://www.cs.nyu.edu/~roweis/data.html`.

## G.I  Variational logistic regression

The optimized lower bound function when the covariance matrix $\mathbf{C}$ is diagonal is as following.

$$\mathcal{L}(\boldsymbol{\mu}, \boldsymbol{\sigma}) = \mathbb{E}_{\mathbf{z} \sim \mathcal{N}(0,\mathbf{I})}[\log l(\boldsymbol{\mu} + \boldsymbol{\sigma} \odot \mathbf{z})] + \frac{1}{2}\sum_{i=1}^{d}\log\frac{\sigma_i^2}{\sigma_i^2 + \mu_i^2},$$

where $l$ is the likelihood function.

The results are shown in Fig. 2 and Fig. 3.

## G.II  Variational Auto-encoder

The results are shown in Fig. 4 and Fig. 5.

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

Figure 2: Convergence rate on variational logistic regression: HFSGVI converges within 3 iterations on small datasets.

Figure 3: Estimated regression coefficients

(a) Convergenve

(b) Reconstruction and Manifold

Figure 4: (a) Convergence rate in terms of epoch. (b) Manifold Learning of generative model when $d_z = 2$: two coordinates of latent variables $\mathbf{z}$ take values that were transformed through the inverse CDF of the Gaussian distribution from equal distance grid on the unit square. $p_{\boldsymbol{\theta}}(\mathbf{x}|\mathbf{z})$ is used to generate the images.

(a) HFSGVI

(b) L-BFGS-SGVI

(c) Ada-SGVI

Figure 5: Reconstruction Comparison