[Reviews · NeurIPS 2015]

Submitted by Assigned_Reviewer_1

This paper presents a scalable second-order stochastic variational inference method for models with normally distributed latent variables. In order to efficiently compute the gradient and Hessian, a representrization trick for general location-scale family is adopted with computation scales quadratically w.r.t the number of latent variables for both gradient and Hessian (in practice with diagonal Gaussian). Furthermore, Hessian-free optimization is used to account for the high dimensionality of the underlying embedded parameters. R-operator technique is used and it enables exact Hessian-vector product computation in Hessian-free optimization. L-BFGS can also be used in place of Hessian-free within the proposed framework. On the theoretical side, the paper provides a variance bound which explains why in some earlier work even only one sample from the variational distribution is sufficient to construct a noisy gradient with low variance. Experimental results are reported on both Bayesian logistic regression and variational autoencoder. The proposed Hessian-free or L-BFGS based methods understandably converge faster and to better local optima.

**Quality**

Overall the paper is of good quality with both practical and theoretical contributions. Deriving bounds is not my expertise, so I couldn't follow all of the proof for Theorem 2 in the supplementary material. But as far as I can understand, the derivation and proof seem sound. The evaluation clearly demonstrates the advantage of the proposed methods.

**Clarity**

This paper seems to be written in a hurry with some parts not clear enough. Some of the descriptions are vague (mostly in the Section 3 and 4) and probably difficult to understand without enough context - I have to interpolate the authors' intention sometimes when reading the paper. I would suggest the authors carefully proof-reading the paper and filling in more context if possible.

Detailed comments:

1. Line 113: Presumably g and H denote the gradient and Hessian of f(z) respectively, but it was never formally defined in the main paper (it is defined in supplementary material).

2. Line 128: leading to an algorithmic -> (based on supplementary material) leading to an algorithmic complexity

3. Line 199: none-the-less -> nonetheless

4. Supplementary material Line 160: lawyer-wise -> layer-wise

5. Supplementary material Line 250: Should be Eq. (10) not Eq. (11).

6. For the second part of the experiment, it is not clear to me why there is running time/objective reported for both train and test. From what I understand, the variational auto-encoder is trained on the training data to learn the parameters for both the encoder and decoder networks with the proposed and competing methods. For test, you feed the test images as the input to the encoder network and reconstruct them from the output of the decoder network (as I understand this is how the images in Figure 2(b) and 3(b) are generated, the authors didn't mention anything about how they are generated). Does the fact that there is also a running time for test mean that the test data is also used to train the model? I hope not, otherwise the reconstruction results would not be reliable. We just need more context to understand the evaluation. If the space is a concern, I would suggest removing some of the introduction to BFGS and some background stuff in Section 3.

**Originality**

The proposed method is built on top of the work on stochastic gradient variational Bayesian (SGVB), which mostly only uses first-order method with adaptive learning rate tuning mechanism (e.g. AdaGrad/RMSProp). Incorporating second-order information isn't something groundbreaking but the paper presents a way to efficiently compute the gradient and Hessian, which is still fairly valuable. Furthermore, Hessian-free and L-BFGS can be applied and benefit from the efficient gradient computation. The theoretical analysis on the variance bound is very exciting work because it provides the theoretical justification for something that has been observed empirically.

**Significance**

The evaluation is reasonably thorough. Given that there isn't new algorithm/modeling innovation in this paper, reporting optimization objective v.s. time is the most convincing way to demonstrate a better optimization strategy.

I do have one comment on the Bayesian logistic regression experiment and would like the authors to investigate and maybe address it in the supplementary material (given that the space seems already a problem for the main paper): Looking at the coefficients in Figure 3 of the supplementary material, the magnitude of DSVI is orders of magnitude larger than the proposed methods for the first two datasets. Is there a particular reason behind this? On the much bigger A9a dataset, all three methods learn coefficients which are in the same scale.

Summary: This paper presents a scalable second-order stochastic variational inference method for models with normally distributed latent variables. It also makes theoretical contribution on the variance bound of the approximation to the mean from samples. The evaluation demonstrates the advantage of the proposed methods on reasonably realistic datasets. There is minor issue with the clarity, but in general the quality is good and I vote for acceptance.

Submitted by Assigned_Reviewer_2

==update after rebuttal== The rebuttal cited some related work I was not previously aware of, particularly Bordes et al. 2009. The authors should discuss relations to these works in the main text

I am not particularly satisfied with the response to theoretical guarantees: The Bordes paper cited is specific to linear SVMs, a quadratic objective, while the variational inference objective is far more complicated (and nonconvex!) In contrast, Robbins-Monro-style algorithms can guarantee convergence to local optima for nonconvex objectives used in variational inference (Bottou 1998, Hoffman et al. 2013).

Re: Bayesian logistic regression being too easy, I did not mean that the problem itself is too easy to merit additional research, but that the datasets chosen are too easy in the sense that 2 of the datasets are too small and all of the algorithms attain such small test errors that it is hard to compare their performance.

I agree with Reviewer 6 that it would be good to have a runtime comparison not based on wallclock time.

After reading the other reviews, it has become more apparent that the NIPS audience would be interested in learning about stochastic second-order methods for variational inference, caveats, notwithstanding, so I have updated my review. ==end of update==

The authors derive efficient formulas for second order derivatives with respect to the variational parameters an expectation of a Lipschitz-differentiable function with respect to a variational Gaussian distribution. They then derive an efficient Hessian-vector multiplication formula and use that to devise a conjugate gradient and L-BFGS algorithm for second-order optimization. In addition, they derive variance bounds for their stochastic gradient estimators that depend only on the Lipschitz constants of the gradients.

The paper is clearly written, and sufficient context is provided to understand its relation to existing work. Both mathematics and experimental methods are clearly presented.

The authors provide valuable technical contributions in Theorems 1 and 2, which decreases the complexity of second order derivative computations compared to naive methods and rigorously bounds the sampling error, respectively. However, the practical effect of these contributions are mitigated. First, the worst-case per-iteration complexity of Algorithm 1 is still O(d^2 d_z^2) (compared to O(d d_z^2) for SGB) because conjugate gradients requires d iterations to be guaranteed to converge. The authors claim to obtain good results by running CG for K < d iterations, but this is not rigorously investigated theoretically or empirically, particularly in the stochastic context. Second, while the variance bound may be of independent interest, it still does not justify a stochastic second-order method, because we have no theorem relating the convergence of second-order methods to the variance of various stochastic estimators.

In the absence of stronger theory, it remains to evaluate the significance of this work by its experimental results. Unfortunately, these are a bit weak. The small Bayesian logistic regression datasets (DukeBreast and Leukemia) are too easy, and all methods achieve very low test error. It is also unclear that any method is the winner on A9a, and confidence intervals are necessary. In addition, comparison to frequentist logistic regression is also in order. The results for the faces autoencoder are also not particularly compelling. The only quantitative results are on the evidence lower bound, not on any test error, which makes it difficult to evaluate the practical benefit of the second order method, since a higher lower bound does not always imply better test performance. Even in terms of optimization performance, the speed benefits of this work only appears to be 2-4x faster than that of Adagrad, but lacks the convergence guarantees of the latter. In addition, it seems that K, the number of CG iterations, can be an important tuning parameter in the algorithm, but the experiments do not investigate different choices of K.
Summary: The authors extend the reparameterization tricks for stochastic gradients used in auto-encoding variational Bayes to second order optimization methods, in particular conjugate gradients and L-BFGS. The chief weaknesses are that (1) stochastic second order methods lack convergence guarantees (as opposed to stochastic gradient methods, which have a rich body of theory), and (2) the empirical comparisons are weak: the Bayesian logistic regression problem is too easy, while the variational auto-encoder problems lack any measure of predictive error. The second order method is shown to attain a larger ELBO in a smaller time, but it is not clear that this improvement in optimization performance translates to any practical benefit, particularly when weighed against the increased complexity and lack of guarantees of the second order method.

Submitted by Assigned_Reviewer_3

The paper extends the stochastic backpropagation algorithm, which is a 1nd order stochastic optimization method for maximizing variational lower bounds over differentiable probabilistic models (typically neural nets) to a 2nd order algorithm that incorporates stochastic second order derivatives over the variational parameters. Such an algorithmic development is certainly very

challenging and the paper goes all the way to firstly derive computationally efficient second order Gaussian identities (see Theorem 1) and then combine stochastic backpropagation with

Hessian-free optimization and limited memory BFGS.

I believe that this is a very mature research work that fully develops the theory of second order stochastic optimization of the Gaussian variational approximation (something that is done for a first time) and at the same time it provides practical implementations of computational efficient algorithms. The experimental comparisons between the 2nd and the previous 1nd order approaches is comprehensive and it shows that the 2nd order methods can speed up optimization.

Specific Comment: Regarding the point "HFSGVI may not behave as fast as Ada in some situations, e.g., when the pixel values of images are sparse", it would be nice if you discuss this further and/or add an illustrative experiment in the supplementary material.
Summary: Second order stochastic backpropagation. An excellent research paper.

Author Feedback
Author rebuttal: We thank the reviewers for their comments. In this response, we would like to address their remarks below.

Reviewer 1
We agree with the suggestion of proofreading and including a more specific description of VAE. Due to space limitations, we will add this to the supplementary material. For comment 6, the time given in our experiment is mainly training time, because the lower bound evaluation on test data takes much less time than during model training. Specifically, test and train lower bounds are evaluated every hundred of SVI iterations. The test data is definitely not used for model training. In terms of the Bayesian logistic regression experiments, the first two datasets are both high-dimensional (7k) and small data size (40). Our 2nd order algorithms can achieve a larger lower bound (relevant to accuracy) quickly but the lower bound does not necessarily indicate sparsity on the coefficients. The VB method can automatically impose sparsity via prior, but this requires longer training. We found running our algorithm longer increased lower bound just a little but induced more sparsity.

Reviewer 2
Thanks for detailed comments.
Q: Stochastic 2nd order methods lack convergence guarantees.
A: The 2nd order algorithms we discuss are function free, with an expectation of the form E[f]. We mainly deal with variational inference E[logf], which is actually equivalent to natural gradient descent (i.e. Hessian is fisher information; as discussed in our survey, LDA is one of the few models using 2nd order information for nice conjugacy). The natural gradient has been thoroughly researched (e.g. DM Blei JMLR2013; S Mukerherjee, Information Theory, IEEE trans 2015; Amari 1998, Neural computation). There is also a substantial amount of work on using stochastic 2nd order algorithms (J Sohl-Dickstein ICML2014, J Ngiam ICML2011, NN Schraudolph AISTAT2007), and theoretical guarantee: 2nd order needs O(d/p) iterations while first order needs O(dc^2/p), where p is desired accuracy and c is matrix conditioner (L Bottou JMLR2009).
Q: Bayesian logistic regression (logreg) is too easy; VAE lacks measurement of the error.
A: Logreg is widely used in applications (e.g. Google Ads) and is a baseline model used to evaluate non-conjugate inference in many VB papers, e.g. DSVI (ICML2014, only logreg), VBI with stochastic search (DM Blei, ICML2012, logreg+LDA), VB message passing (TP Minka, NIPS2011, logreg+softmaxreg). For logreg datasets, we have used the same datasets as DSVI and DSVI has compared with frequentist logreg. In terms of reporting measurement error, we mentioned in section 3.2 that the negative lower bound of VAE is equivalent to the cross entropy error or squared error for binary and Gaussian input respectively, regardless of a constant or multipliable scalar difference. Thus test lower bound is equivalent to predictive reconstruction error.
Q: Choice of CG Iteration K.
A: We mentioned the worst case of CG for solving linear systems in our paper. The preconditioned CG is used and theoretically the quantitative relation between the iteration K and relative tolerance e is e < exp(-2K/sqrt{c}) (JR Shewchuk, 1994). The negative exponential convergence rate implies why practically CG often converges within few iterations. Also the inequality indicates that the conditioner c can be nearly as large as O(K^2). For the choice of K, we mentioned we tested different Ks (5 to 25 in fact). But we selected the best results with minimum K. We did not include the result because it wasn't our focus, but we can add it to the supplementary material.
Q: Variance bound doesn't justify stochastic 2nd order method.
A: The variance bound and its corollary are novel and guarantee any estimator through the sample mean, thus justifying our 2nd order SVI. Since both the gradient and the Hessian are in the expectation and estimated stochastically, a good estimator with fewer samples and lower variance will help the gradient based algorithm.

Reviewer 3
Thanks for the useful comments. If the image pixels are sparse, like in MNIST, the input nodes will contain many 0s, thus leaving the connected weights untrained (dropout effect). The software implementation has a fast sparse matrix operation and storage. But the Hessian matrix is less likely to be sparse. HFSGVI has no significant improvement w.r.t running time. In addition, we have a comparison with MNIST in the supplementary material looking at the amount of training data evaluated.

Reviewer 4
Thanks for your comments.

Reviewer 5
Thanks for your comments.

Reviewer 6
Thanks for your comments. Our paper is the first to propose a 2nd order optimization framework for a non-conjugate generative model. Besides algorithmic novelty, we also included a novel variance bound and sample mean convergence rate. The code available from relevant authors and our algorithms are all implemented in Matlab. Our paper also used the same datasets and framework as referenced papers.